# Macrophage *Sult2b1* promotes pathological neovascularization in age-related macular degeneration

Yafang Wang[1], Yang Liu[1], Yan Wang[2], Yidong Wu[1], Zhixuan Chen[1], Feng Wang[7], Xiaoling Wan[1,3] (iD), Fenghua Wang[1,3,4,5,6], Xiaodong Sun[1,3,4,5,6]

**Disordered immune responses and cholesterol metabolism have been implicated in age-related macular degeneration (AMD), the leading cause of blindness in elderly individuals. SULT2B1, the key enzyme of sterol sulfonation, plays important roles in inflammation and cholesterol metabolism. However, the role and underlying mechanism of SULT2B1 in AMD have not been investigated thus far. Here, we report that SULT2B1 is specifically expressed in macrophages in choroidal neovascularization lesions. *Sutl2b1* deficiency significantly reduced leakage areas and inhibited pathological angiogenesis by inhibiting M2 macrophage activation in vivo and in vitro. Mechanistically, loss of *Sult2b1* activated LXRs and subsequently increased ABCA1 and ABCG1 (ABCA1/G1)-mediated cholesterol efflux from M2 macrophages. LXR inhibition (GSK2033 treatment) in *Sult2b1*$^{-/-}$ macrophages reversed M2 polarization and decreased intracellular cholesterol capacity to promote pathological angiogenesis. In contrast to SULT2B1, STS, an enzyme of sterol desulfonation, protected against choroidal neovascularization development by activating LXR–ABCA1/G1 signalling to block M2 polarization. Collectively, these data reveal a cholesterol metabolism axis related to macrophage polarization in neovascular AMD.**

## Introduction

Age-related macular degeneration (AMD), the leading cause of blindness among elderly people (1), occurs in two main end-stage forms: non-neovascular (dry) AMD and neovascular (wet) AMD (nAMD). Wet AMD, which accounts for ~90% of cases of severe vision loss (2, 3, 4), is characterized by the abnormal extension of blood vessels into the retina, termed choroidal neovascularization (CNV). Given the key role of VEGF in the pathology of nAMD, anti-VEGF therapy is currently the gold standard of disease management

(3), but treatment responses vary, and long-term anti-VEGF treatment may cause severe retinal adverse events (5, 6, 7, 8). Thus, there is an urgent need to explore alternative approaches to treat nAMD.

The accumulation of lipid-rich deposits, namely, drusen, between the retinal pigment epithelium (RPE) and Bruch's membrane is a pathological hallmark in the early and intermediate stages of AMD (2, 9). Drusen participates in nAMD progression, resulting in immune cell recruitment and activation (10). Of the infiltrating innate immune cells in nAMD lesions, we and other groups have demonstrated that macrophages are the main cell type, indicating that they play an essential role in the pathogenesis of this disease (11, 12, 13, 14). In response to various tissue-derived or environmental stimuli, macrophages can undergo polarization to the pro-inflammatory (M1) or proangiogenic (M2) phenotype (14, 15). Compared with M1 macrophages, M2 macrophages are the predominant participant in the pathological angiogenesis of nAMD (16, 17, 18). Modulation of this phenotype change may be a novel therapeutic strategy for the treatment of nAMD.

Recent studies have shown that cholesterol metabolism is a key regulator of macrophage activation (19, 20). Given the high-cholesterol content of drusen in subretinal lesions, it is crucial to better understand the role of abnormal cholesterol metabolism in macrophage polarization in nAMD. Sulfotransferase family 2b member 1 (SULT2B1) is the key enzyme in the sulfate conjugation of 3β-hydroxysteroids such as cholesterol, oxysterol, and pregnenolone (21, 22, 23), a process that ultimately affects cholesterol metabolism. The desulfation process is catalysed by steroid sulfatase (STS). Our previous study showed that *Sult2b1* deficiency promoted pro-inflammatory macrophage polarization and exacerbated ischaemic stroke in the brain (24), the occurrence of which is positively associated with increased cholesterol (25). However, the potential role and underlying mechanism of *Sult2b1* in nAMD remain unclear.

In this study, we show that *Sult2b1* deficiency attenuates CNV development by inhibiting proangiogenic M2 macrophage polarization

[1]Department of Ophthalmology, Shanghai General Hospital (Shanghai First People's Hospital), Shanghai Jiao Tong University School of Medicine, Shanghai, China [2]Medical Research Center, Peking University Third Hospital, Beijing, China [3]Shanghai Key Laboratory of Ocular Fundus Diseases, Shanghai, China [4]Shanghai Engineering Center for Visual Science and Photomedicine, Shanghai, China [5]National Clinical Research Center for Eye Diseases, Shanghai, China [6]Shanghai Engineering Center for Precise Diagnosis and Treatment of Eye Diseases, Shanghai, China [7]Shanghai Institute of Immunology, Department of Immunology and Microbiology, State Key Laboratory of Oncogenes and Related Genes, Shanghai Jiao Tong University School of Medicine, Shanghai, China

Correspondence: wangfeng16@sjtu.edu.cn; shaolin.72@163.com; shretina@sjtu.edu.cn

in vivo and in vitro. Moreover, loss of *Sult2b1* activated liver X receptors (LXR)-ABCA1 and ABCG1 (ABCA1/G1) signalling for efficient cholesterol efflux, consequently preventing M2 macrophage polarization. In contrast to the effects of *Sult2b1* deficiency, knockout of *Sts* inactivated LXR–ABCA1/G1 signalling and promoted both M2 macrophage polarization and pathologic neovascularization. Our findings highlight the importance of SULT2B1 and STS in regulating macrophage polarization and cholesterol metabolism in the pathologic neovascularization of nAMD, which might provide a potential therapeutic strategy for nAMD.

# Results

### *Sult2b1* deficiency decreases the CNV lesion area in vivo

To better understand the role of *Sult2b1* in nAMD in vivo, we measured *Sult2b1* mRNA levels in RPE–choroid tissue in a laser-induced mouse model of nAMD. During CNV lesion development, *Sult2b1* mRNA levels were significantly higher (CNV3d, $P < 0.001$; CNV5d, $P < 0.01$; CNV7d, $P < 0.05$) in the laser-induced CNV group than in the control group (Fig S1), indicating that *Sult2b1* may play an important role in CNV development.

Then, we performed laser photocoagulation of the ocular fundus in WT- and *Sult2b1*-deficient (*Sult2b1*[-/-]) mice to assess whether *Sult2b1* deficiency alters CNV, as represented by leakage or neovascularization areas. The fundus fluorescein angiography (FFA) results showed that 7 d after laser photocoagulation, the leakage area of the laser-induced lesions was significantly smaller ($P < 0.0001$) in *Sult2b1*[-/-] mice than in WT mice (Fig 1A and B). The neovascularization area (labelled with isolectin) was also significantly smaller ($P < 0.001$) in *Sult2b1*[-/-] mice than in WT mice (Fig 1C and D). Consistently, the mRNA levels of proangiogenic growth markers (*Vegfa*, *Pdgfb*, and *Tek*), an endothelial cell marker (*Tek*), and profibrotic factors (*Tgfb1*, *Pdgfb*, and *Fgf2*) were lower in the CNV lesions of *Sult2b1*[-/-] mice than in those of WT mice (Fig 1E).

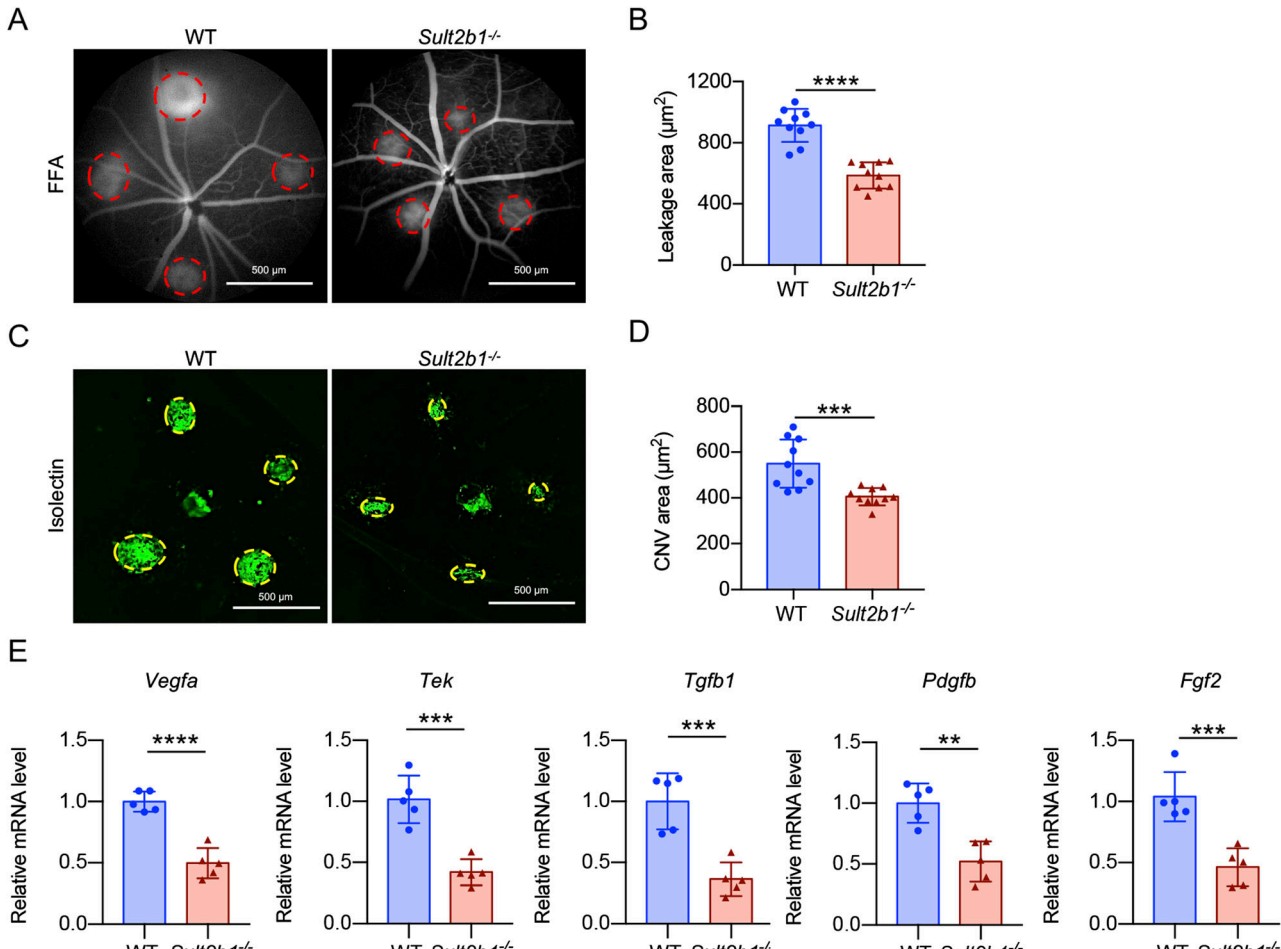

**Figure 1.  *Sult2b1* deficiency attenuates CNV in a mouse model of nAMD.**
**(A, B)** Representative images of fundus fluorescein angiography (A) and quantification of the leakage area (red circle, (B)) in WT and *Sult2b1*[-/-] mice (n = 10 per group). Scale bar, 500 $\mu$m. **(C, D)** Representative isolectin (green) staining (C) of RPE–choroid flat mounts and quantification of the CNV area (yellow circle, (D)) in WT and *Sult2b1*[-/-] mice (n = 10 per group). Scale bar, 500 $\mu$m. **(E)** Relative mRNA levels of proangiogenic growth markers (*Vegfa*, *Pdgfb*, and *Tek*), an endothelial cell marker (*Tek*), and profibrotic factors (*Tgfb1*, *Pdgfb*, and *Fgf2*) in the RPE–choroid flat mounts of WT and *Sult2b1*[-/-] mice (n = 5 per group). Data information: in (B, D, E), data are presented as the mean ± SD. **$P < 0.01$, ***$P < 0.001$, ****$P < 0.0001$ (unpaired two-tailed *t* test).

Together, these results suggest that *Sult2b1* facilitates the progression of laser-induced CNV.

## SULT2B1+ macrophages are required for CNV progression in vivo

To determine which cell type is responsible for CNV development, we first examined the expression pattern of SULT2B1 in CNV lesions. Interestingly, SULT2B1 was specifically expressed in macrophages (F4/80+) in the CNV areas of WT mice but not of *Sult2b1*-/- mice (Fig 2A). Moreover, we noted less macrophage infiltration in the CNV lesions of *Sult2b1*-/- mice than in those of WT mice according to F4/80 staining of RPE–choroid flat mounts (Fig 2B and C), suggesting that SULT2B1+ macrophages are involved in the laser-induced CNV.

To verify the hypothesis, we depleted macrophages in WT and *Sult2b1*-/- mice by intraperitoneal injection (IP) of a Csf1r and c-Kit inhibitor (pexidartinib, PLX3397) that selectively induces macrophage death (26). After a continuous 7-d treatment with PLX3397, we observed a significant reduction ($P < 0.01$) in macrophages in the spleen compared with the untreated group (Fig 2D and E). After PLX3397 treatment, there was no significant difference in CNV area between *Sult2b1*-/- and WT mice (Fig 2F–I). Therefore, these results suggest that SULT2B1+ macrophages contribute to laser-induced CNV.

## *Sult2b1* drives M2 polarization to promote pathological angiogenesis

Given that alternatively activated M2 macrophages can promote pathological angiogenesis (16, 17), we investigated whether *Sult2b1* expression affects M2 macrophage polarization in vivo and in vitro. Immunofluorescence analysis of CNV lesions showed that the percentage of M2 macrophages (indicated by YM1 or ARG1 positivity) was lower in the CNV lesions of *Sult2b1*-/- mice than in those of WT mice (Fig 3A–D). Then, we applied BMDMs in vitro and employed IL-4 to induce M2 polarization (27). Consistently, the SULT2B1 protein level in BMDMs showed a marked increase after IL-4 stimulation (Figs 3E and S2), which indicated that *Sult2b1* might play a role in M2 polarization. Moreover, YM1 and ARG1 protein levels were significantly lower in *Sult2b1*-/- BMDMs than in WT BMDMs after the induction of M2 polarization (Fig 3F). Taken together, these results indicate that *Sult2b1* deletion inhibits differentiation to M2 macrophages.

To test our hypothesis that *Sult2b1* is related to M2-mediated vessel formation in CNV lesions, we measured the angiogenic activity of M2 BMDMs harvested from *Sult2b1*-/- and WT mice. To this end, we incubated C166 cells and HUVECs with the conditioned medium of WT or *Sult2b1*-/- M2 BMDMs and performed tube formation assays to assess vessel formation. The M2 BMDMs from *Sult2b1*-/- mice formed significantly fewer vessels in the three-dimensional (3D) matrix than those from WT mice (Fig 3G and J), consistent with the in vivo CNV results. Thus, *Sult2b1* deficiency negatively affects the ability of macrophages to regulate pathological angiogenesis by inhibiting M2 polarization.

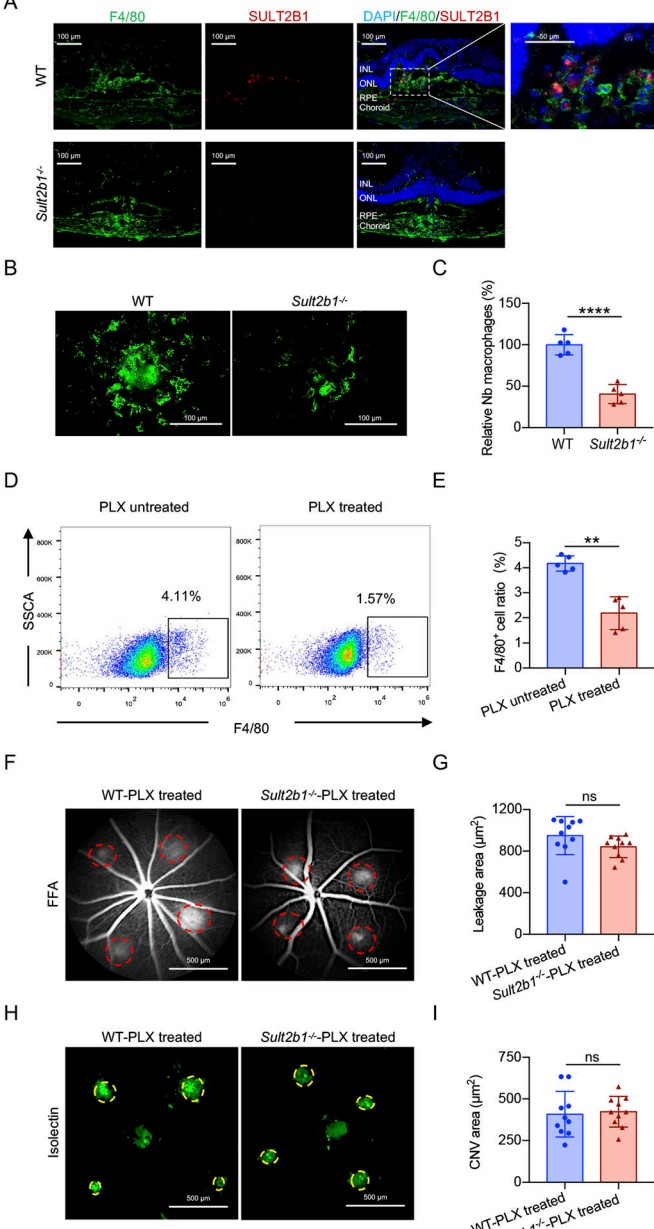

**Figure 2. SULT2B1-expressing macrophages are involved in laser-induced CNV.**
**(A)** Immunostaining of F4/80 (green) and SULT2B1 (red) in retinal sections from WT and *Sult2b1*-/- mice 7 d after laser application (n = 5 per group). Blue indicates DAPI. INL, inner nuclear layer; ONL, outer nuclear layer; RPE, retinal pigment epithelium. **(B, C)** Immunostaining of F4/80 (green) in the flat-mounted RPE–choroid complexes (B) and quantification of macrophage infiltration in the CNV lesions (C) of WT and *Sult2b1*-/- mice (n = 5 per group). Scale bars, 100 μm. **(D, E)** Flow cytometry dot plots (D) and a bar graph (E) of F4/80+ cells collected from the spleens of mice treated with or without PLX3397 (n = 5 per group). **(F, G)** Representative images from the fundus fluorescein angiography analysis (F) and quantification (G) of the leakage area (red circle) of PLX3397 (a Csf1r and c-Kit inhibitor, 16 mg/kg, IP)-treated WT and *Sult2b1*-/- mice (n = 10 per group). Scale bars, 500 μm. **(H, I)** Representative isolectin staining (green) in RPE–choroid flat mounts (H) and quantification (I) of the CNV area (yellow circle) in WT and *Sult2b1*-/- after PLX3397 treatment (n = 10 per group). Scale bars, 500 μm. Data information: in (C, E, G, I), data are presented as the mean ± SD. **$P < 0.01$, ****$P < 0.0001$, no significance (ns) (unpaired two-tailed t test).

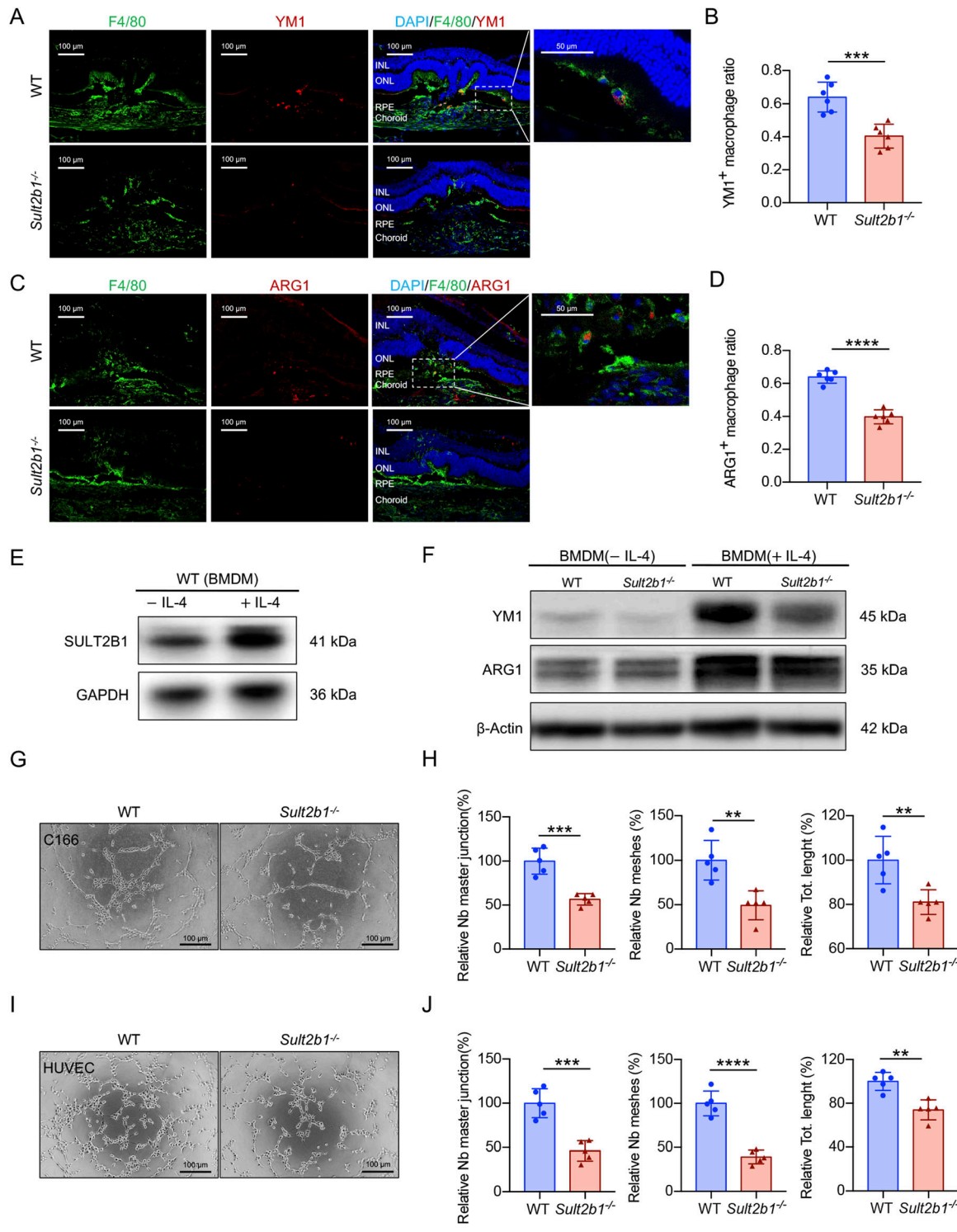

**Figure 3. *Sult2b1* drives macrophages towards M2 polarization and promotes neovascularization.**
**(A, B)** Representative images of double staining for F4/80 (green) and YM1 (red) (A) and quantification (B) of YM1-positive macrophages in CNV lesions of WT and *Sult2b1⁻/⁻* mice (n = 6 per group). **(C, D)** Representative images of double staining for F4/80 (green) and ARG1 (red) (C) and quantification (D) of ARG1-positive macrophages in CNV lesions of WT and *Sult2b1⁻/⁻* mice (n = 6 per group). Blue indicates DAPI. INL, inner nuclear layer; ONL, outer nuclear layer; RPE, retinal pigment epithelium. **(E)** Western blot analysis of SULT2B1 in WT BMDMs treated with or without IL-4. GAPDH was used as a loading control. **(F)** Western blot analysis of YM1 and ARG1 in BMDMs from WT and *Sult2b1⁻/⁻* mice treated with or without IL-4. β-Actin was used as a loading control. **(G)** Representative images of tube formation by C166 cells cultured with conditioned medium from WT or *Sult2b1⁻/⁻* M2 BMDMs. Scale bar, 100 μm. **(H)** Quantification of tube formation by C166 cells (n = 5). **(I)** Representative images of tube formation by HUVECs cultured with a conditioned medium from WT or *Sult2b1⁻/⁻* M2 BMDMs. Scale bar, 100 μm. **(I, J)** Quantification of tube formation by HUVECs (I) (n = 5).
Data information: in (B, D, H, J), data are presented as the mean ± SD. **P < 0.01, ***P < 0.001, ****P < 0.0001 (unpaired two-tailed *t* test).
Source data are available for this figure.

## Cholesterol efflux mediated by LXR–ABCA1/G1 signalling is increased in *Sult2b1⁻/⁻* macrophages

Having pinpointed that *Sult2b1* is involved in M2 polarization, we next sought to explore the mechanism by which *Sult2b1* skews macrophage polarization towards the M2 phenotype during CNV development. LXR, a key regulator of cholesterol homoeostasis, can be antagonized by *Sult2b1* through the sulfation of oxysterols (23). In addition, the induction of efficient cholesterol efflux from senescent macrophages via treatment with an LXR agonist can regulate M2 polarization and pathological angiogenesis (28, 29). Hence, we speculated that *Sult2b1* regulates pathological angiogenesis through LXR signalling. To test this hypothesis, we carried out immunofluorescence analyses of macrophages from *Sult2b1⁻/⁻* CNV lesions and found that LXRα and LXRβ protein levels were elevated (Fig 4A and B). Furthermore, the proteins encoded by LXR target genes, including the cholesterol reverse transporters *ATP-binding cassette subfamily members A1* (*ABCA1*) and *G1* (*ABCG1*), were at significantly higher levels in macrophages from *Sult2b1⁻/⁻* CNV lesions than in those from WT lesions, as shown by immunofluorescence (Fig 4C and D). Consistently, LXRα, LXRβ, ABCA1, and ABCG1 protein levels were increased in *Sult2b1⁻/⁻* M2 BMDMs (Fig 4E). Therefore, signalling through LXR is suppressed by *Sult2b1* deficiency in macrophages.

Furthermore, we determined the total intracellular cholesterol content in macrophages by fluorometry and found that cholesterol accumulation was attenuated in BMDMs of *Sult2b1⁻/⁻* mice (Fig 4F). Moreover, the cholesterol efflux rate of *Sult2b1⁻/⁻* BMDMs was higher than that of WT BMDMs (Fig 4G). These results demonstrate that the loss of *Sult2b1* leads to the activation of LXR–ABCA1/G1 signalling, which subsequently accelerates cholesterol transport out of macrophages.

## LXR inhibition eliminates the capacity of *Sult2b1⁻/⁻* macrophages to restrain pathological angiogenesis

Thus, to evaluate the role of LXRs in pathological angiogenesis in the context of *Sult2b1⁻/⁻* macrophages, we investigated whether the application of LXR antagonist (inhibiting LXRα and LXRβ, GSK2033) reversed the phenotype of *Sult2b1⁻/⁻* macrophages in vivo and in vitro. After generating CNV lesions by laser photocoagulation, *Sult2b1⁻/⁻* and WT mice were treated by intravitreal injection of GSK2033. The results showed that the expression levels of the LXRs and their targets ABCA1 and ABCG1 were successfully reduced by GSK2033 treatment (Fig 5A–D). However, there was no significant difference in their expression between *Sult2b1⁻/⁻* and WT macrophages treated with the LXR antagonist (Fig 5A–D). Furthermore, GSK2033 significantly increased the leakage area (*P* < 0.0001) and CNV lesion size (*P* < 0.0001), regardless of *Sult2b1* deficiency (Fig 5E and F), indicating that LXR inhibition could protect against pathological angiogenesis. The leakage area and CNV size in treated *Sult2b1⁻/⁻* mice were larger than those in untreated *Sult2b1⁻/⁻* mice. Interestingly, CNV lesion sizes were comparable between the *Sult2b1⁻/⁻* and WT mice upon treatment with GSK2033 (Fig 5E and F), suggesting that the effects of *Sult2b1* deficiency depend mainly on LXRs.

Moreover, GSK2033 treatment resulted in decreased protein levels of LXRα, LXRβ, ABCA1, and ABCG1 in WT BMDMs (Fig 6A and B) and increased the protein levels of YM1 and ARG1 in WT M2 BMDMs (Fig 6C and D), indicating that LXR inhibition could prevent M2 polarization regardless of *Sult2b1* expression status. Notably, the expression levels of LXRs, LXR targets, and M2 markers in *Sult2b1⁻/⁻* macrophages treated with the LXR antagonist did not significantly differ from those in treated WT macrophages (Fig 6A–D). We also examined the effect of the LXR antagonist on the tube-formation ability of WT and *Sult2b1⁻/⁻* macrophages in vitro. The result showed that there was no significant difference in forming vessels between WT and *Sult2b1⁻/⁻* macrophages treated by GSK2033 (Figs 6E and F and S3A and B), which were consistent with those observed in vivo (Fig 5E and F). Further analysis of cholesterol content and cholesterol efflux revealed that treatment with the LXR antagonist reversed both the decrease in cholesterol content and the increase in cholesterol efflux in *Sult2b1*-deficient macrophages (Fig 6G and H). It may suggest that the effect of *Sult2b1* on cholesterol efflux capacity relies on LXRs. Overall, our data suggest that the anti-angiogenic characteristics of *Sult2b1⁻/⁻* macrophages are most likely attributable to the activation of LXRs.

## *Sts* deficiency aggravates CNV by promoting M2 macrophage differentiation

In contrast to *Sult2b1*, *Sts* catalyses the conversion of sulfated oxysterol to oxysterol (30, 31). Therefore, we sought to determine whether the loss of *Sts* plays an opposing role in the pathological angiogenesis of CNV. To accomplish this, we performed laser photocoagulation in WT- and *Sts*-deficient (*Sts⁻/⁻*) mice (Fig S4) and examined CNV lesions and infiltrating macrophages. STS and SULT2B1 were specifically expressed in macrophages within the CNV lesion (Fig 7A). Strikingly, *Sts⁻/⁻* mice had larger CNV lesions after laser injury than the WT mice (Fig 7B–E). The mRNA levels of proangiogenic growth markers (*Vegfa*, *Pdgfb*, and *Tek*), an endothelial cell marker (*Tek*), and profibrotic factors (*Tgfb1*and *Pdgfb*) were higher in the CNV lesions of *Sts⁻/⁻* mice than in those of WT mice (Fig 7F). Moreover, the number of infiltrating macrophages was significantly increased in the CNV lesions of *Sts⁻/⁻* mice, as determined by analysis of RPE–choroid flat mounts (Fig S5A and B). The percentage of M2 macrophages (indicated by YM1 or ARG1 positivity) in *Sts⁻/⁻* CNV lesions was higher than that in WT lesions (Fig S5C–F). In addition, the expression levels of LXRs and their targets ABCA1 and ABCG1 in macrophages were lower in CNV lesions from *Sts⁻/⁻* mice than in those from WT mice (Fig S6A–D). These results demonstrate that *Sts* protects against CNV development by activating LXR–ABCA1/G1 signalling to block M2 macrophage differentiation, indicating that its role is opposite to that of *Sult2b1*.

# Discussion

The findings of our present study demonstrate that *Sult2b1* plays an important role in CNV development by promoting M2 polarization. Specifically, we found that *Sult2b1* deficiency leads to enhanced cholesterol efflux via activation of oxysterol-derived LXR signalling

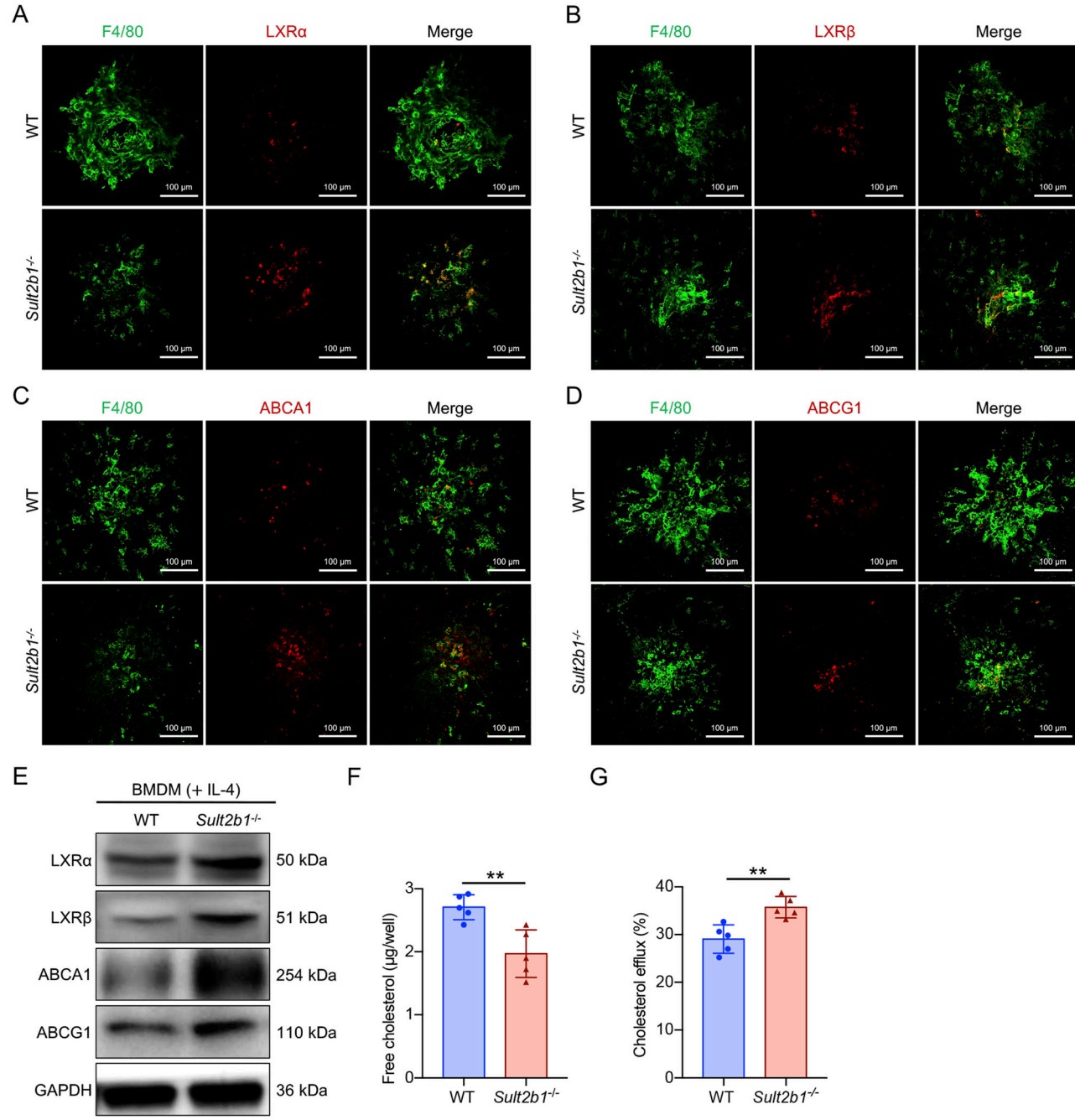

**Figure 4.** *Sult2b1* deficiency leads to LXR activation and ABCA1/G1-mediated cholesterol efflux in macrophages.
**(A, B, C, D)** Representative fluorescence images of macrophages (F4/80⁺) (green) in flat-mounted RPE–choroid complexes from WT or *Sult2b1⁻/⁻* mice; the images show staining for (A) LXRα (red), (B) LXRβ (red), (C) ABCA1 (red), and (D) ABCG1 (red). Scale bar, 100 μm. **(E)** Western blot analysis of LXRα, LXRβ, ABCA1, and ABCG1 in M2 BMDMs from WT and *Sult2b1⁻/⁻* mice. GAPDH was used as a loading control. **(F)** Intracellular cholesterol content in WT and *Sult2b1⁻/⁻* M2 BMDMs (n = 5 per group). **(G)** Cholesterol efflux in WT and *Sult2b1⁻/⁻* M2 BMDMs (n = 5 per group). Data information: in (F, G), data are presented as the mean ± SD. **$P < 0.01$ (unpaired two-tailed $t$ test). Source data are available for this figure.

and that treatment with an LXR antagonist reverses these changes, resulting in decreased cholesterol efflux and increased cholesterol accumulation in the *Sult2b1⁻/⁻* macrophage. Conversely, *Sts*, which converts sulfated oxysterols to oxysterols, has a protective role against pathological neovascularization. These results shed light on the involvement of the cholesterol metabolism axis in macrophage

polarization in nAMD, which may serve as a promising new therapeutic strategy in nAMD.

SULT2B1 has been previously shown to be expressed in multiple cell types, including hepatocytes, lymphocytes, and endothelial cells, in different tissues (32, 33). Here, we showed that SULT2B1 is specifically expressed in the infiltrating macrophages of CNV

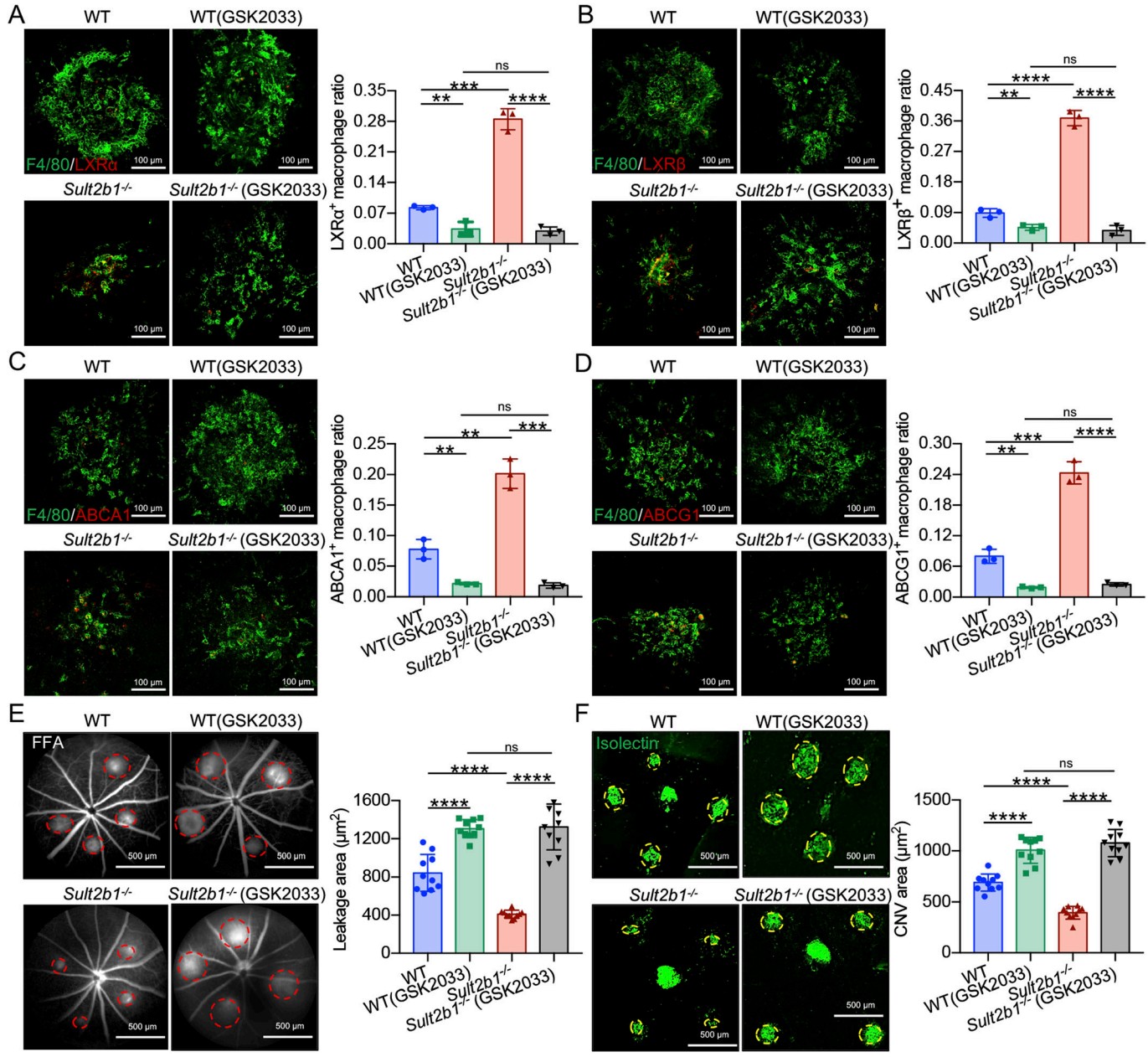

**Figure 5. LXR antagonist treatment decreases the capacity of *Sult2b1*⁻/⁻ macrophages to regulate CNV in vivo.**
**(A, B, C, D)** Macrophages (F4/80⁺, green) in CNV lesions of WT and *Sult2b1*⁻/⁻ mice with GSK2033 (the LXR antagonist, 500 ng/μl, IVI) were stained for (A) LXRα (red), (B) LXRβ (red), (C) ABCA1 (red), and (D) ABCG1 (red) (n = 5 per group). Scale bar, 100 μm. **(E)** Representative images of fundus fluorescein angiography and quantification of the leakage area (red circle) in WT and *Sult2b1*⁻/⁻ mice (n = 10 per group) with or without GSK2033 treatment. **(F)** Representative images of isolectin (green) staining and quantification of CNV area (yellow circle) in RPE–choroid flat mounts from WT and *Sult2b1*⁻/⁻ mice (n = 10 per group) with or without GSK2033 treatment. Scale bar, 500 μm. Data information: in (A B, C, D, E, F), data are presented as the mean ± SD. **P < 0.01, ***P < 0.001, ****P < 0.0001, no significance (unpaired two-tailed t test).

lesions. Our previous work assessing BMDM differentiation in a mouse model of ischaemic stroke suggested that *Sutl2b1* inhibited pro-inflammatory (M1) macrophage polarization ([24]). In this study, we found that *Sutl2b1* promoted proangiogenic (M2) macrophage polarization in a laser-induced CNV mouse model. Collectively, our data indicated that the effect of *Sult2b1* on macrophage polarization and function depends on the pathological context.

Oxysterols are the natural ligands of the LXR nuclear receptors and play crucial roles in cholesterol homoeostasis ([31], [34]). SULT2B1-mediated sulfation of oxysterols and STS-mediated desulfation of sulfated oxysterols are opposing processes that can inactivate and activate LXRs, respectively. Previous studies suggest pivotal roles for SULT2B1/STS-mediated oxysterol-derived LXR signalling in the cholesterol metabolism of the

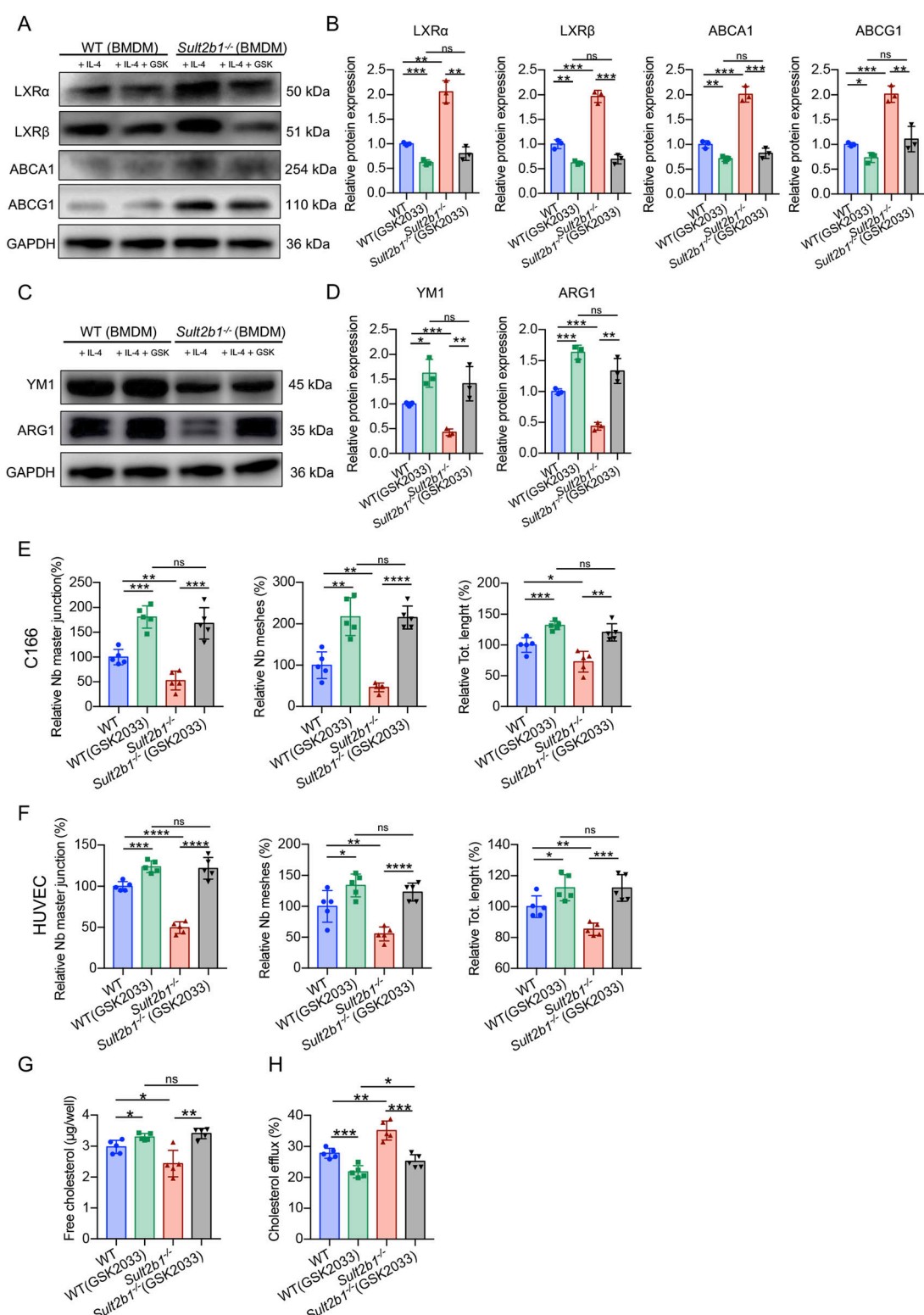

**Figure 6. LXR antagonist treatment reverses the capacity of *Sult2b1⁻/⁻* macrophages to regulate neovascularization in vitro.**
**(A)** Western blot analysis of LXRα, LXRβ, ABCA1, and ABCG1 levels in WT and *Sult2b1⁻/⁻* M2 BMDMs treated with or without GSK2033. **(B)** Relative protein expression of LXRα, LXRβ, ABCA1, and ABCG1 (n = 3). **(C)** Western blot analysis of YM1 and ARG1 levels in WT and *Sult2b1⁻/⁻* BMDMs with or without GSK2033 treatment. **(D)** Relative protein expression of YM1 and ARG1(n = 3). **(E)** Tube formation analysis of C166 cells cultured with a conditioned medium from BMDMs (treated with or without GSK2033) (n = 5). **(F)** Tube formation analysis of HUVECs cultured with conditioned medium from BMDMs (treated with or without GSK2033) (n = 5). **(G)** The intracellular cholesterol

liver but of the eye (30, 35). In our study, knockout of *Sult2b1* activated LXR–ABCA1/G1-mediated cholesterol efflux, subsequently inhibiting M2 polarization and attenuating pathologic neovascularization. However, *Sts* deficiency had the opposite effect on CNV development. In conclusion, our study is the first to explore the roles and mechanisms of *Sult2b1* and *Sts* in nAMD.

ABCA1/G1 facilitates cholesterol efflux to high-density lipoprotein particles for formation and maturation, thereby maintaining cholesterol homoeostasis (36). ABCA1 and ABCG1 are highly expressed in macrophages, especially in response to excess cholesterol accumulation (37, 38). Efficient cholesterol efflux mediated by the cholesterol transporters ABCA1/G1 in senescent macrophages can protect against pathologic angiogenesis (28). One study showed that conditional knockout of *Abca1* and *Abcg1* in macrophages leads to age-associated cholesterol-rich deposits underneath the retina, suggesting that impaired cholesterol efflux in macrophages initiates the development of AMD (39). Our study showed that the expression levels of ABCA1 and ABCG1 were increased in *Sult2b1*$^{-/-}$ macrophages, wherein these proteins promoted cholesterol transport and reduced cholesterol accumulation in CNV lesions. Treatment of *Sult2b1*$^{-/-}$ macrophages with an LXR antagonist reversed all these phenotypes, resulting in decreased cholesterol efflux capacity and increased cholesterol accumulation. In short, our result suggests that the anti-angiogenic characteristics of *Sult2b1*$^{-/-}$ macrophages are most likely dependent on LXR–ABCA1/G1 signalling.

In summary, the current study demonstrates that *Sult2b1* deficiency attenuates CNV by inhibiting M2 macrophage polarization and promoting cholesterol efflux via the LXR–ABCA1/G1 pathway. Our findings shed light on the regulation of macrophage polarization and pathologic neovascularization through cholesterol metabolism.

# Materials and Methods

### Animals

Animal procedures were approved by the Shanghai Jiao Tong University Institutional Review Board. All animal experiments conformed to the Association for Research in Vision and Ophthalmology (ARVO) Statement for the Use of Animals in Ophthalmic and Vision Research. The C57BL/6J mice were provided by the Laboratory Animal Center of Shanghai General Hospital. The *Sult2b1*$^{-/-}$ mice were provided by Shanghai Institute of Immunology, as described in a previous study (40). The *Sts*$^{-/-}$ mice were generated by deleting 91 bp in exon 2 of the *Sts* gene as described in Fig S4. The target ending with NGG near exon 2 was designed and cut under the action of CRISPR/Cas9. The sgRNA1 sequence of sgRNA1 was GAGGATGATGTCAGCGAGCGTGG, and the sgRNA2 sequence of sgRNA2 was AGCCGAGGTCACCGATGCCCAGG. Primer sequences for

genotyping are presented in Table S1. Male mice aged between 6 and 8 wk and ~20 g were included in the experiment.

### Laser-induced CNV

Mice were anaesthetized with an intraperitoneal injection of 1% pentobarbital sodium (0.1 ml/10 g body weight) after the application of tropicamide (Santen) for pupil dilatation. CNV lesions were induced as described in a previous study (11). Briefly, four laser spots were administered around the optic nerve head by an argon laser (120 mW in intensity, 50 $\mu$m in size, and 100 ms in duration) (OcuLight Infrared Laser System 810 nm; Iridex Corp.) in each eye. CNV lesion area was determined by FFA and immunofluorescence in RPE–choroid flat mounts on day 7 after injury.

### Intravitreal injection

Mice were anaesthetized as described above. After laser treatment, the mice were immediately divided into two groups for intravitreal injection. For local anaesthesia, benoxinate HCl 0.4% drops (Santen) were administered. Subsequently, 2 $\mu$l of either GSK2033 (LXR antagonist, 500 ng/$\mu$l) or vehicle (PBS) was delivered into the vitreous cavity using a microsyringe (Hamilton). The operation was conducted under a dissecting microscope (Olympus) in a sterilized environment. After the operation, 0.5% levofloxacin eye drops (Santen) were applied to prevent infection.

### FFA

Mice were anaesthetized, and their pupils were dilated as described above. Then, fluorescein sodium (1%, 0.1 ml/kg; Alcon) was injected intraperitoneally, and late-phase angiograms were performed 5 min after the injection. Fluorescein angiography (FA) was performed by a Micron IV System (Phoenix Research Laboratories). The area of each CNV lesion with leakage was quantified using ImageJ software (National Institutes of Health).

### Macrophage depletion

Pexidartinib (16 mg/kg, PLX3397; MedChemExpress) was injected intraperitoneally twice daily for seven consecutive days before laser photocoagulation. No obvious behavioural or health problems were observed in animals injected with PLX3397. Splenocytes were collected and stained with F4/80 (123108; BioLegend) on day 7 after PLX3397 treatment. Because cessation of PLX3397 treatment induces macrophage regeneration (41), we administered PLX3397 for 7 d after laser injury.

### Cell culture

HUVECs, from ATCC were grown in humidified air (37°C, 5% $CO_2$) in an endothelial cell medium (1001; Solarbio) supplemented with 5%

---

content of WT and *Sult2b1*$^{-/-}$ M2 BMDMs treated with or without GSK2033 (n = 5). **(H)** Cholesterol efflux in WT and *Sult2b1*$^{-/-}$ BMDMs with or without GSK2033 treatment (n = 5). Data information: in (B, D, E, F, G, H), data are presented as the mean ± SD. *$P < 0.05$, **$P < 0.01$, ***$P < 0.001$, ****$P < 0.0001$, no significance (ns) (unpaired two-tailed *t* test). Source data are available for this figure.

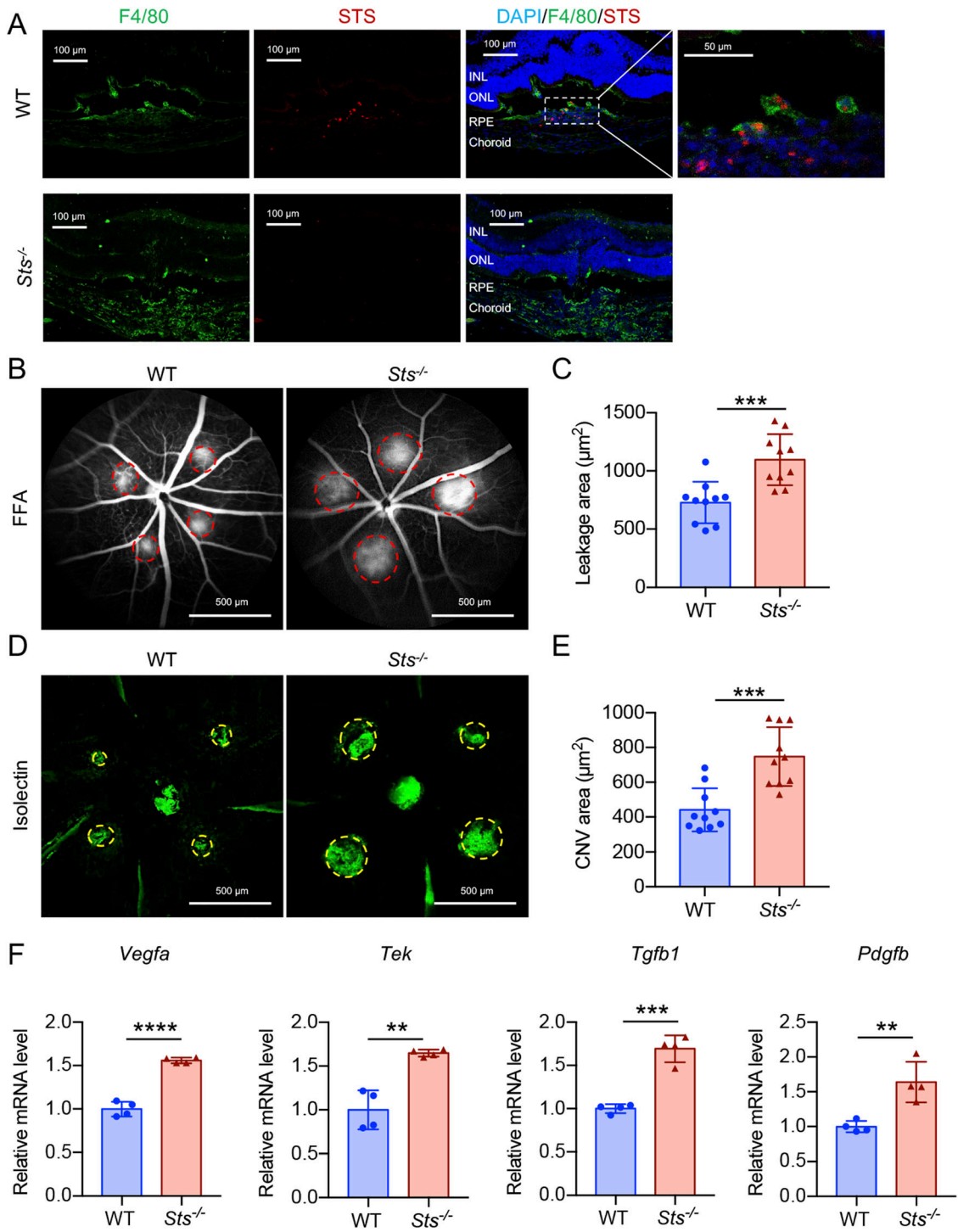

**Figure 7.  *Sts* deficiency promotes pathological angiogenesis.**
**(A)** Cryosections of WT and *Sts*⁻/⁻ mouse CNV lesions stained for F4/80 (green) and STS (red). Blue indicates DAPI. INL, inner nuclear layer; ONL, outer nuclear layer; RPE, retinal pigment epithelium. **(B, C)** Representative images of fundus fluorescein angiography and leakage area (red circle) (B) and quantification of the leakage area (C) in WT and *Sts*⁻/⁻ mice (n = 10 per group). **(D, E)** Representative images of isolectin (green) staining and CNV area (yellow circle) in RPE–choroid flat mounts (D) and quantification of the CNV area (E) in WT and *Sts*⁻/⁻ mice (n = 10 per group). **(F)** Relative mRNA levels of proangiogenic growth markers (*Vegfa*, *Pdgfb*, and *Tek*), an endothelial cell marker (*Tek*), and profibrotic factors (*Tgfb1* and *Pdgfb*) in the RPE-choroid flat mounts of WT and *Sts*⁻/⁻ mice (n = 4 per group). Data information: In (C, E, F), data are presented as the mean ± SD. **P < 0.01, ***P < 0.001, ****P < 0.0001 (unpaired two-tailed *t* test).

FBS, 1% penicillin–streptomycin antibiotics and 1% endothelial cell growth supplement. C166 cells (from ATCC) and BMDMs were cultured in DMEM/high glucose (SH30243.01; Hyclone) with 10% FBS and 1% penicillin–streptomycin antibiotics.

## BMDM polarization

Bone marrow cells were harvested from the femur and tibia and then differentiated with 10 ng/ml recombinant mouse M-CSF (PMC2044; Thermo Fisher Scientific) for 7 d [24]. For M2 polarization, the macrophages were stimulated with 10 ng/ml recombinant mouse IL-4 (574302; BioLegend) [42] for 24 h. The conditioned medium and cells were collected separately for the following use.

## Endothelial cell tube formation assay

Matrigel matrix (354234; Corning) was added to a 96-well plate (50 $\mu$l/well) and allowed to polymerize for 30 min at 37°C. Then, 1 × 10$^4$ C166 cells or HUVECs and conditioned medium from BMDMs (described above) were mixed and seeded on the Matrigel matrix. After 6 h of incubation at 37°C, tube formation was photographed using a light microscope at 10× magnification. Automated analysis was performed with the ImageJ software (National Institutes of Health, Bethesda, MD, USA). Three parameters (number [Nb] of master junctions, Nb of meshes, and total length) were taken for the quantification experiments.

## Cholesterol quantification assay

The free-cholesterol content of M2 BMDMs was measured using the cholesterol/cholesteryl ester quantitation assay kit (ab65359; Abcam). In brief, the dried lipids (extracted from 1 × 10$^6$ M2 BMDMs) were dissolved in 200 $\mu$l assay buffer (supplied in the kit), and then incubated with a cholesterol probe/cholesterol enzyme mix solution (supplied in the kit) at 37°C for 1 h protected from light. Finally, the fluorescence was detected by a fluorescent microplate reader (Ex/Em = 535/587 nm). The free-cholesterol content was quantified using a cholesterol standard curve.

## Cholesterol efflux assay

The cholesterol efflux was measured using the commercial cholesterol efflux assay kit (cell-based) (ab196985; Abcam). In brief, M2 BMDMs (1 × 10$^5$) were incubated with a labelling reagent for 1 h and then with the Equilibration Buffer overnight. The cells were treated with the desired cholesterol acceptor (serum) for 6 h in an incubator (37°C, 5% CO$_2$). The fluorescence intensity of the media and cell lysates was measured via a fluorescent microplate reader (Ex/Em = 482/515 nm). The cholesterol efflux was calculated by dividing the fluorescence intensity of the media by the total fluorescence intensity of the cell lysate and media.

## Immunofluorescence

Eyeballs were harvested from mice that were transcardially perfused with 4% paraformaldehyde and then prepared for RPE–choroid flat mounts or cryosections. The antibodies used for staining were anti-isolectin (FL-1201, 1:500; Vector Labs), anti-F4/80 (ab6640, 1:500; Abcam), anti-SULT2B1 (ab254616, 1:500; Abcam), anti-YM1 (1404, 1:500; Stem Cell), anti-ARG1 (16001-1-AP, 1:500; Proteintech), anti-LXR$\alpha$ (ab176323, 1:100; Abcam), anti-LXR$\beta$ (ab28479, 1:200; Abcam), anti-ABCA1 (ab18180, 1:200; Abcam), anti-ABCG1 (ab52617, 1:100; Abcam), and anti-STS (17870-1-AP, 1:500; Proteintech). Images were visualized by a fluorescence microscope (Olympus). ImageJ (National Institutes of Health, Bethesda, MD, USA) was used for analysis and quantification.

## RNA extraction and real-time PCR

Total RNA was extracted from RPE–choroidal tissues (with CNV lesions) using a total RNA extraction kit (Tiangen). RNA quantity and quality were assessed with a NanoDrop spectrophotometer (Thermo Fisher Scientific). cDNA was synthesized with a cDNA synthesis kit (Takara Bio), and gene expression was analysed by real-time PCR (Applied Biosystems). Primer sequences are presented in Table S2. The mRNA levels of the target genes were quantitatively normalized against those of *Gapdh* using the ΔΔCT method.

## Western blot analysis

Harvested BMDMs described above were homogenized with lysis buffer. The samples were resolved by 10% SDS–PAGE and transferred to PVDF membranes. After blocking with 5% nonfat dry milk in TBST (TBS containing Tween-20) for 1 h at room temperature, the membranes were incubated at 4°C overnight with primary antibodies against GAPDH (10494-1-AP, 1:1,000; Proteintech), $\beta$-actin (3779, 1:1,000; Prosci), YM1 (1404, 1:1,000; Stem Cell), ARG1 (16001-1-AP, 1:5,000; Proteintech), SULT2B1 (ab254616, 1:1,000; Abcam), LXR$\alpha$ (ab176323, 1:1,000; Abcam), LXR$\beta$ (ab28479, 1:1,000; Abcam), ABCA1 (ab18180, 1:500; Abcam) or ABCG1 (ab52617, 1:1,000; Abcam). The membranes were washed three times with TBST and then incubated with the corresponding horseradish peroxidase-conjugated secondary antibodies for 1 h at room temperature. Finally, the membranes were washed three times with TBST, and the bands were visualized by chemiluminescence using a molecular imaging system (Amersham Imager 600; GE Healthcare). GAPDH or $\beta$-actin was used as loading control. The experiment was replicated three times using independent biological samples.

## Statistical analysis

Data are shown as the mean ± SD. Each experiment was repeated three times. Differences among groups were analysed using unpaired two-tailed *t* test (Prism v7.0; Graph-Pad Software, Inc). *P* < 0.05 was set as the criterion for significance.

# Data Availability

The datasets generated during and/or analysed during the current study are available from the corresponding author on reasonable request.

# Supplementary Information

# Acknowledgements

This work was supported by the National Natural Science Foundation of China (81970845, 82000906, 82230055, and 82071852).

## Author Contributions

Y Wang: data curation, formal analysis, methodology, and writing—original draft.
Y Liu: formal analysis and validation.
Y Wang: formal analysis and validation.
Y Wu: formal analysis and investigation.
Z Chen: formal analysis and investigation.
F Wang: conceptualization, data curation, validation, investigation, project administration, and writing—review and editing.
X Wan: conceptualization, data curation, and writing—original draft, review, and editing.
F Wang: project administration.
X Sun: project administration.

## Conflict of Interest Statement

The authors declare that they have no conflict of interest.

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
