## [Reviewer comments · Life Science Alliance]

Life Science Alliance

Macrophage Sult2b1 promotes pathological neovascularization in age-related macular degeneration

Yafang Wang, Yang Liu, Yan Wang, Yidong Wu, Zhixuan Chen, Feng Wang, Fenghua Wang, Xiao-Dong Sun and Xiaoling Wan
DOI: <https://doi.org/10.26508/lsa.202302020>

Corresponding author(s): Dr. Xiaoling Wan (Shanghai General Hospital)

Review Timeline:

Submission Date:	2023-03-01
Editorial Decision:	2023-05-11
Revision Received:	2023-07-02
Editorial Decision:	2023-07-24
Revision Received:	2023-07-28
Accepted:	2023-07-31

Scientific Editor: Novella Guidi

Transaction Report:

May 11, 2023

Re: Life Science Alliance manuscript #LSA-2023-02020-T

Dr. Xiaoling Wan
Shanghai General Hospital
CHINA

Dear Dr. Wan,

Thank you for submitting your manuscript entitled "Macrophage Sult2b1 promotes pathological neovascularization in age-related macular degeneration" to Life Science Alliance. The manuscript was assessed by expert reviewers, whose comments are appended to this letter. We invite you to submit a revised manuscript addressing the Reviewer comments.

Thank you for this interesting contribution to Life Science Alliance. We are looking forward to receiving your revised manuscript.

Sincerely,

B. MANUSCRIPT ORGANIZATION AND FORMATTING:

Reviewer #1 (Comments to the Authors (Required)):

This study reveals a possible role of cholesterol metabolism in CNV pathogenesis. The study is generally well done. However, it helps to resolve a few issues below:

1. It is perhaps understandable why cholesterol transport in macrophage is related to CNV in human patients because drusen deposits consist of lipids and cholesterol. However, it is harder to understand why CNV induced by laser has anything to do with cholesterol. Could the author explain why laser induced CNV is a good model for cholesterol's role in CNV in human patients?
2. Laser-induced CNV is known to be highly variable. The data presented in this study is much less variable than expected. Could the authors show all the raw data that support the quantitation for Figure 1B and 1C (the raw data can be in one file)?
3. The authors proposed a central role of LXR in CNV pathogenesis. If this is the case, LXR agonist should suppress CNV in WT animals according to the model proposed in this study (since LXR antagonist does the opposite). Such pharmacological data would significantly strengthen this study.
4. This enzyme SULT2B1 is too ubiquitous to be a good therapeutic target in human. Even if a particular protein or enzyme is involved in a complex pathway, it does not mean that this protein or enzyme plays an important role in the human disease related to that pathway or is a good therapeutic target.

Reviewer #2 (Comments to the Authors (Required)):

This study uncovers the target of immunological and cholesterol metabolic activity balance associated in age-related macular degeneration (AMD), the leading cause of blindness in elderly individuals. This study reveals Sult2b1, an enzyme expressed in macrophage as the key target to cause choroidal neovascularization (CNV) lesions. In CNV, M2 macrophages are the predominant participant to affect the cholesterol metabolism due to the presence of Sult2b1 in macrophages. The Sult2b1 deficient mice demonstrated to show the activated LXR signaling via the ABCA1 and ABCG1 targets and decreased the CNV lesion. Interestingly, this study provided a proof-of-concept to attenuate the pathological angiogenesis in AMD-CNV by utilizing a small molecule drug, GSK2033 which activates the LXR signaling causing the efflux of cholesterol and inhibiting the M2 polarization.

1. In the 'Material and Methods' section.
 - a. Source of C166 and HUVEC cells used in the study needs to be included.
 - b. Methods involving the Cholesterol measurement and efflux assay needs to be described in detail like sample preparation type, methods, the instruments, and the parameters which were utilized to measure the cholesterol levels and how the data were interpreted.
2. Figures and legends:
 - a. Figure S1
 - i. Y-axis should be labeled as Log2 fold change.
 - ii. Legend doesn't have explanation for CNV-3, 5 & 7-days.
 - b. Figure 1
 - i. From S1, the most mRNA levels of Sult2b1 from RPE-Choroid tissue was noted be highly significant at 3-day. What was the rationale behind choosing 'seven days' after laser photocoagulation as a data point to interpret the leakage and CNV area (A-D)?
 - ii. Y-axis should be labeled as Log2 fold change.
 - c. Figure 2
 - i. Scale bar in panel A is missing.
 - ii. Naming of layers such as the RPE, Choroid and the retinal layer in the 3rd column within panel A-having the DAPI staining with markers is essential for easy navigation to the readers.
 - iii. The zoomed areas in the 3rd column within panel A is unclear. Those boxes probably need to be separated out as 4th column within Panel A to get a clear view of the co-localization pattern of F4/80 with Sult2b1.

d. Figure 3

- i. Scale bar in panel A, C, G and I is missing.
- ii. Naming of layers such as the RPE, Choroid and the retinal layer in the 3rd column within panel A & C-having the DAPI staining with markers is essential for easy navigation to the readers.
- iii. The zoomed areas showing the co-localization of macrophages (F4/80) with M2 macrophages (YM1 or ARG1) in the 3rd column within panel A & C is essential to understand the expression pattern clearly.
- iv. Labeling [such as WT and WT(IL-4)] in panel E is confusing and not clear. It could be re-labelled as BMDM-WT (-IL4) and BMDM-WT (+IL4). Additionally, the role of IL4 inclining to the macrophages to polarize to M2 needs to be stated.
- v. Labeling pattern in Panel F is also confusing and not clear. It could be re-labelled as +IL4 and -IL4.
- vi. What was the rationale for the use of two different cell lines (C166 and HUVEC) by the authors to show the tube formation assays and assess the vessel formation?

e. Figure 4

- i. Was the immunofluorescence analysis done on RPE-choroid flat mounts? If so, this must be stated.
- ii. The molecular weight of the protein targets analyzed in Panel E needs to be indicated.
- iii. Since the western blotting in Panel E was done using M2-BMDMs derived from the WT and Sult2b1^{-/-} mice, I'm curious to know if the authors performed IL4 treatment before taking samples for western blot experiment? Also, it would be ideal to add an additional label tag of BMDM on top of WT and Sult2b1^{-/-}

f. Figure 6

- i. In PANEL-A, the ABCA1 blot is not clear. Bands are not visible in the WT lane for this target.
- ii. It would be interesting if the authors could quantitate the data of panel A and B by taking densitometric measurements and plotting them as graphs.

g. Figure 7

- i. Naming of layers such as the RPE, Choroid and the retinal layer in the 3rd column within panel A-having the DAPI staining with markers is essential for easy navigation to the readers.
- ii. It would be interesting to show the relative mRNA levels of proangiogenic growth markers (Vegfa, pdgfb, and Tek), endothelial marker (Tek), and profibrotic factors (Tgfb1, Pdgfb, and Fgf2) from the WT and STS^{-/-} mice (as shown in Fig 1 between WT and sult2b1^{-/-} mice)

h. Figure S5

- i. Retinal layers need to be labeled.

i. Table S1

- i. Did the authors design a Forward and Reverse primer for genotyping the Sult2b1^{-/-} and Sts^{-/-} mice? If so, this needs to be specified in the table.

Dear Editors,

We really appreciated the editors and reviewers for your time and constructive comments on our manuscript entitled “**Macrophage Sult2b1 promotes pathological neovascularization in age-related macular degeneration**” (LSA-2023-02020-T). Those comments are valuable and helpful for revising our paper. According to all the suggestions and questions, we have revised the whole manuscript. We do our best to answer the questions, the point-by-point responses to the comments are attached below.

Sincerely,

Xiaoling Wan

Department of Ophthalmology, Shanghai General Hospital

School of Medicine, Shanghai Jiao Tong University

100 Haining Road, Shanghai 200080, China

Reviewer #1 (Comments to the Authors (Required)):

This study reveals a possible role of cholesterol metabolism in CNV pathogenesis. The study is generally well done. However, it helps to resolve a few issues below:

Response: We sincerely appreciate your positive evaluation of our work and constructive suggestions. Below we addressed and clarified your comments in a point-by-point manner, and revised our manuscript accordingly.

1. It is perhaps understandable why cholesterol transport in macrophage is related to CNV in human patients because drusen deposits consist of lipids and cholesterol. However, it is harder to understand why CNV induced by laser has anything to do with cholesterol. Could the author explain why laser induced CNV is a good model for cholesterol's role in CNV in human patients?

Response: Thanks for your constructive suggestion.

The laser-induced CNV mouse model was first developed by the Campochiaro group in 1998 (1). At present, it has been a crucial mainstay model for nAMD research. The angiogenesis resulting from laser induction greatly mimics the main characteristics of CNV in human patients (2, 3).

As you mentioned above, one of the limitations in this acute mouse model is that the subretinal RPE deposits (drusen) are absent or not crucial, which is unlike in CNV of patients. To explore the role of cholesterol metabolism in CNV *in vivo*, researchers still tried to use this laser-induced CNV mouse model and did a series of studies. For example, high-fat/cholesterol-fed mice showed an impaired ability to regulate pathological angiogenesis in a laser-induced CNV mouse model (4). Macrophage specific deletion of cholesterol transporter *Abcal* (ATP binding cassette transporter) and *Scd2* (stearoyl-CoA desaturase-2) promoted pathological angiogenesis in a laser-induced CNV mouse model (4, 5). These studies demonstrated that cholesterol metabolism played a vital role in the laser-induced CNV mouse model. We were inspired by these studies and cited them. In addition, the laser-induced CNV mouse

model has been widely used in evaluating therapies targeting components of lipid metabolism for nAMD, such as omega-3 long-chain polyunsaturated fatty acids (LCPUFAs) (6-9), apolipoprotein A-I (ApoA1) and ApoAI-binding protein (AIBP) (10), apolipoprotein M (ApoM) (11) and so on. Given these studies, we applied this model to investigate the role and underlying mechanism of SULT2B1 in nAMD *in vivo*.

In conclusion, although the laser-induced CNV mouse model is not perfect, it contributes to the understanding of the pathogenesis of nAMD, as well as to the development of nAMD therapies.

2. Laser-induced CNV is known to be highly variable. The data presented in this study is much less variable than expected. Could the authors show all the raw data that support the quantitation for Figure 1B and 1C (the raw data can be in one file)?

Response: Thanks for your great suggestion. Yes, laser-induced CNV is known to be highly variable. To rule out this factor, the laser-induced CNV model in our study was constructed by a skilled operator (*Yafang Wang*). Actually, our lab has long been dedicated to basic and translational studies on pathological neovascularization in AMD (12-16) and has accumulated rich experience in establishing the laser-induced CNV animal model. All experiments were independently performed at least three times to ensure repeatable results.

As you suggested, we showed all the raw data in Figure R1. The leakage areas and CNV areas were quantified using ImageJ software (National Institutes of Health, Bethesda, MD, USA).

Figure R1

Figure R1 The raw data of Figure 1A–D.

(A–B) The images of fundus fluorescein angiography (A) and quantification of the leakage area (red circle, B) in WT and *Sult2b1*^{-/-} mice (n=10 per group). Scale bar, 500 μ m.

(C–D) Isolectin (green) immunostaining (C) of RPE-choroid flat mounts and quantification of CNV area (yellow circle, D) in WT and *Sult2b1*^{-/-} mice (n=10 per group). Scale bar, 500 μ m.

3. The authors proposed a central role of LXR in CNV pathogenesis. If this is the case, LXR agonist should suppress CNV in WT animals according to the model proposed in this study (since LXR antagonist does the opposite). Such pharmacological data would significantly strengthen this study.

Response: Thank you for raising this very important and insightful point. Your hypothesis is correct. Actually, the central role of LXR in CNV pathogenesis was reported by Sene et al. (4). They treated old mice with LXR agonist (T0901317) and found a significant and dose-dependent reduction in CNV compared to vehicle-treated old mice, demonstrating the important role of LXR in regulating pathological vascularization (Figure R2) (4). When we tried to explore the mechanism by which *Sult2b1* skews macrophage polarization during CNV development, we were inspired by this excellent work and cited it.

To make our study more solid, as you suggested, LXR agonist (T0901317, MCE) was treated in WT mice by intravitreal injection. The results showed that the expression levels of the LXRs were successfully activated by T0901317 treatment (Figure R3 E and F). Importantly, the CNV areas were significantly reduced after T0901317 treatment (Figure R3 A–D), which was consistent with the previous study.

Figure R2

Figure R2 Representative data captured from Figure 4 in a previous study (4).

(H–I) (H) Representative CNV images in vehicle and LXR agonist-treated old mice and quantification (I) of CNV (white circle) volume show a dose-dependent reduction of endothelial cell proliferation. Values are expressed as mean + SE. Statistically significant difference, * $p < 0.05$, ** $p < 0.01$, compared to vehicle treatment.

Figure R3

Figure R3 LXR agonist treatment alleviates CNV in a mouse model of nAMD.

(A–B) Representative images of fundus fluorescein angiography (A) and quantification of the leakage area (red circle, B) in WT mice (n=10 per group). Scale bar, 500 µm.

(C–D) Representative isolectin (green) staining (C) of RPE-choroid flat mounts and quantification of CNV area (yellow circle, D) in WT mice (n=10 per group). Scale bar, 500 µm.

(E–F) Representative fluorescence images of macrophages (F4/80⁺) (green) in CNV lesions from WT mice; the images show staining for LXRα (red) (E) and LXRβ (red) (F). Scale bar, 100 µm.

Data information: In (B, D), data are presented as the mean ± SD. ****P < 0.0001 (Student's unpaired two-tailed t-test).

4. This enzyme SULT2B1 is too ubiquitous to be a good therapeutic target in human. Even if a particular protein or enzyme is involved in a complex pathway, it does not mean that this protein or enzyme plays an important role in the human disease related to that pathway or is a good therapeutic target.

Response: Yes. We fully agree with your opinion. In the present study, we aim to investigate the role and underlying mechanism of SULT2B1 in nAMD. As a result, we demonstrated a cholesterol metabolism axis (SULT2B1/STS-LXR-ABCA1/G1) related to macrophage polarization in neovascular AMD. Whether the enzyme SULT2B1 is an intervenable target in nAMD remains further studied. As you discussed, SULT2B1 seems an unlikely therapeutic target. Other groups focused on the targeting of LXR as a therapeutic intervention (17, 18), such as LXR agonist (4) and anti-microRNA antisense oligonucleotide (anti-miR33) (19).

Reviewer #2 (Comments to the Authors (Required)):

This study uncovers the target of immunological and cholesterol metabolic activity balance associated in age-related macular degeneration (AMD), the leading cause of blindness in elderly individuals. This study reveals Sult2b1, an enzyme expressed in macrophage as the key target to cause choroidal neovascularization (CNV) lesions. In CNV, M2 macrophages are the predominant participant to affect the cholesterol metabolism due to the presence of Sult2b1 in macrophages. The Sult2b1 deficient mice demonstrated to show the activated LXR signaling via the ABCA1 and ABCG1 targets and decreased the CNV lesion. Interestingly, this study provided a proof-of-concept to attenuate the pathological angiogenesis in AMD-CNV by utilizing a small molecule drug, GSK2033 which activates the LXR signaling causing the efflux of cholesterol and inhibiting the M2 polarization.

Response: Thanks for your careful review and invaluable suggestions. Below we addressed and clarified all comments in the point-by-point responses, and revised our manuscript accordingly.

1. In the 'Material and Methods' section.

- a. Source of C166 and HUVEC cells used in the study needs to be included.
- b. Methods involving the Cholesterol measurement and efflux assay needs to be described in detail like sample preparation type, methods, the instruments, and the parameters which were utilized to measure the cholesterol levels and how the data were interpreted.

Response: Thank you for pointing them out. We have added the information.

Revised:

‘Human umbilical vein endothelial cells (HUVECs, from ATCC) were grown in humidified air (37°C, 5% CO₂) in endothelial cell medium (Solarbio, 1001) supplemented with 5% fetal bovine serum (FBS), 1% penicillin–streptomycin antibiotics and 1% endothelial cell growth supplement. C166 cells (from ATCC) and bone marrow-derived macrophages were cultured in DMEM/High glucose (Hyclone, SH30243.01) with 10% FBS and 1% penicillin–streptomycin antibiotics.’

‘Cholesterol quantification assay

The free cholesterol content of M2 BMDMs was measured using the Cholesterol/Cholesteryl Ester Quantitation Assay Kit (Abcam, ab65359). In brief, the dried lipids (extracted from 1×10^6 M2 BMDMs) were dissolved in 200 μ l Assay Buffer (supplied in the Kit), and then incubated with Cholesterol Probe/Cholesterol Enzyme Mix solution (supplied in the Kit) at 37°C for one hour protected from light. Finally, the fluorescence was detected by a fluorescent microplate reader (Ex/Em = 535/587 nm). The free cholesterol content was quantified using a cholesterol standard curve.’

‘Cholesterol efflux assay

The cholesterol efflux was measured using the commercial Cholesterol Efflux Assay Kit (Cell-Based) (Abcam, ab196985). In brief, M2 BMDMs (1×10^5) were incubated

with Labeling Reagent for one hour and then with the Equilibration Buffer overnight. The cells were treated with the desired cholesterol acceptor (serum) for 6 hours in an incubator (37°C, 5% CO₂). The fluorescence intensity of the media and cell lysates was measured via a fluorescent microplate reader (Ex/Em = 482/515 nm). The cholesterol efflux was calculated by dividing the fluorescence intensity of the media by the total fluorescence intensity of the cell lysate and media.'

2. Figures and legends:

Response: Thank you for pointing them out.

a. Figure S1

i. Y-axis should be labeled as Log₂ fold change.

ii. Legend doesn't have explanation for CNV-3, 5 & 7-days.

Response:

i. For the real-time PCR (RT-PCR) assay, the fold-change in the expression of each target mRNA relative to *Gapdh* was calculated using the CT ($2^{-\Delta\Delta CT}$) method (<https://toptipbio.com/delta-delta-ct-pcr/>). Each experiment was performed in triplicate. Fold change = $2^{-\Delta\Delta CT}$; $\Delta Ct = Ct(Sult2b1) - Ct(Gapdh)$; $\Delta\Delta Ct = \Delta Ct(CNV-3d/5d/7d) - \Delta Ct(CNV-0d)$. Therefore, the Y-axis is represented as the relative mRNA level (fold change). The mRNA level of CNV-0d was standardized at 1. To make it more unambiguous, 'mRNA level (fold change)' has been changed to 'relative mRNA level'.

Figure S1

Figure S1 *Sult2b1* mRNA levels during CNV progression.

Quantitative mRNA analysis of *Sult2b1* levels in RPE-choroid flat mounts during CNV development (n=3). CNV-0d indicates the time point before laser injury. CNV-3/5/7d indicates 3/5/7 days after laser injury. Data information: Data are presented as the mean \pm SD. *P < 0.05, **P < 0.01, ***P < 0.001 (Student's unpaired two-tailed t-test).

ii. CNV-0d indicates the time point before laser injury. CNV-3/5/7d indicates 3/5/7 days after laser injury. We have added. Thanks!

b. Figure 1

i. From S1, the most mRNA levels of *Sult2b1* from RPE-Choroid tissue was noted be highly significant at 3-day. What was the rationale behind choosing 'seven days' after laser photocoagulation as a data point to interpret the leakage and CNV area (A-D)?

ii. Y-axis should be labeled as Log2 fold change.

Response:

i. To better understand the role of *Sult2b1* in nAMD *in vivo*, we measured *Sult2b1* mRNA levels in RPE-choroid tissue in a laser-induced mouse model of nAMD. During CNV lesion development, *Sult2b1* mRNA levels were significantly higher (CNV3d, $P < 0.001$; CNV5d, $P < 0.01$; CNV7d, $P < 0.05$) in the laser-induced CNV group than in the untreated group (Figure S1). However, there was no significant difference between CNV3d, CNV5d, and CNV7d.

The optimal time point to assess CNV formation (CNV area) is typically at day 7 (2, 3). We selected this time point of seven days after laser photocoagulation based on the characters of the CNV mouse model rather than the expression level of the target gene.

ii. As described above (a), the $2^{-\Delta\Delta CT}$ method was used for relative quantification. Fold change = $2^{-\Delta\Delta CT}$. To make it more unambiguous, ‘mRNA level (fold change)’ has been changed to ‘relative mRNA level’.

Figure 1

c. Figure 2

i. Scale bar in panel A is missing.

ii. Naming of layers such as the RPE, Choroid and the retinal layer in the 3rd column within panel A-having the DAPI staining with markers is essential for easy navigation to the readers.

iii. The zoomed areas in the 3rd column within panel A is unclear. Those boxes probably need to be separated out as 4th column within Panel A to get a clear view of the co-localization pattern of F4/80 with *Sult2b1*.

Response:

i. Scale bar in panel A was added. Thanks.

ii. Naming of layers (INL, ONL, RPE, and Choroid) in the 3rd column were added. Thanks.

iii. The zoomed area in the 3rd column within panel A has been separated out as 4th column. Thanks.

Figure 2

(A) Immunostaining of F4/80 (green) and SULT2B1 (red) in retinal sections from WT and *Sult2b1*^{-/-} mice 7 days after laser application (n=5 per group). Blue indicates DAPI. INL, inner nuclear layer; ONL, outer nuclear layer; RPE, retinal pigment epithelium.

d. Figure 3

- i. Scale bar in panel A, C, G and I is missing.
- ii. Naming of layers such as the RPE, Choroid and the retinal layer in the 3rd column within panel A & C-having the DAPI staining with markers is essential for easy navigation to the readers.
- iii. The zoomed areas showing the co-localization of macrophages (F4/80) with M2 macrophages (YM1 or ARG1) in the 3rd column within panel A & C is essential to understand the expression pattern clearly.
- iv. Labeling [such as WT and WT(IL-4)] in panel E is confusing and not clear. It could be re-labelled as BMDM-WT (-IL4) and BMDM-WT (+IL4). Additionally, the role of IL4 inclining to the macrophages to polarize to M2 needs to be stated.
- v. Labeling pattern in Panel F is also confusing and not clear. It could be re-labelled as +IL4 and -IL4.
- vi. What was the rationale for the use of two different cell lines (C166 and HUVEC) by the authors to show the tube formation assays and assess the vessel formation?

Response:

- i. We added scale bars in panel A, C, G and I. Thanks!
- ii. Naming of layers (INL, ONL, RPE, and Choroid) in the 3rd column within panel A & C were added. Thanks.
- iii. The zoomed areas in the 3rd column within panel A & C have been separated out as 4th column. Thanks.

Figure 3

(A) Representative images of double staining for F4/80 (green) and YM1 (red) in CNV lesions of WT and *Sult2b1*^{-/-} mice.

(C) Representative images of double staining for F4/80 (green) and ARG1 (red) in CNV lesions of WT and *Sult2b1*^{-/-} mice.

Blue indicates DAPI. INL, inner nuclear layer; ONL, outer nuclear layer; RPE, retinal pigment epithelium.

iv. We have re-labelled in the figure.

The role of IL-4 inclining to the macrophages to polarize to M2 has been added in the manuscript. Thanks.

‘Then, we applied bone marrow-derived macrophages (BMDMs) in vitro and employed IL-4 to induce M2 polarization (20).’

v. We have re-labelled in the figure.

Figure 3

vi. C166 is an endothelial cell line that was isolated from mouse embryo. HUVECs are umbilical vein endothelial cells from human. These two cell lines are commonly used in vascular diseases. In the present study, we used two endothelial cell lines of different species to do tube formation assays. Consistently, the M2 BMDMs from *Sult2b1*^{-/-} mice formed significantly fewer vessels in the three-dimensional (3D) matrix than those from WT mice.

e. Figure 4

i. Was the immunofluorescence analysis done on RPE-choroid flat mounts? If so, this must be stated.

ii. The molecular weight of the protein targets analyzed in Panel E needs to be indicated.

iii. Since the western blotting in Panel E was done using M2-BMDMs derived from the WT and *Sult2b1*^{-/-} mice, I'm curious to know if the authors performed IL4 treatment before taking samples for western blot experiment? Also, it would be ideal to add an additional label tag of BMDM on top of WT and *Sult2b1*^{-/-}.

Response:

i. Yes. We have added in the figure legend. Thanks.

‘(A–D) Representative fluorescence images of macrophages (F4/80⁺) (green) in flat-mounted RPE-choroid complexes from WT or *Sult2b1*^{-/-} mice.’

ii. The molecular weight of the protein targets analyzed in Panel E has been indicated.

iii. The samples in Figure 4E were treated with IL-4. We have added an additional label tag of BMDM on top of WT and *Sult2b1*^{-/-}.

Figure 4E

f. Figure 6

i. In PANEL-A, the ABCA1 blot is not clear. Bands are not visible in the WT lane for this target.

ii. It would be interesting if the authors could quantitate the data of panel A and B by taking densitometric measurements and plotting them as graphs.

Response:

i. In PANEL-A, we have changed a clear picture. Thanks.

ii. As you suggested, we have quantitated the data of panel A and B.

Figure 6 A-D

(A) Western blot analysis of LXRα, LXRβ, ABCA1, and ABCG1 levels in WT and *Sult2b1*^{-/-} M2 BMDMs treated with or without GSK2033.

(B) Relative protein expression of LXRα, LXRβ, ABCA1, and ABCG1 (n=3).

(C) Western blot analysis of YM1 and ARG1 levels in WT and *Sult2b1*^{-/-} BMDMs with or without GSK2033 treatment.

(D) Relative protein expression of YM1 and ARG1 (n=3).

g. Figure 7

i. Naming of layers such as the RPE, Choroid and the retinal layer in the 3rd column within panel A-having the DAPI staining with markers is essential for easy navigation to the readers.

ii. It would be interesting to show the relative mRNA levels of proangiogenic growth markers (Vegfa, pdgfb, and Tek), endothelial marker (Tek), and profibrotic factors (Tgfb1, Pdgfb, and Fgf2) from the WT and STS^{-/-} mice (as shown in Fig 1 between WT and *sult2b1*^{-/-} mice)

Response:

i. Naming of layers (INL, ONL, RPE, and Choroid) in the 3rd column within panel A were added. Thanks.

Figure 7A

(A) Cryosections of WT and *Sts*^{-/-} mouse CNV lesions stained for F4/80 (green) and STS (red). Blue indicates DAPI. INL, inner nuclear layer; ONL, outer nuclear layer; RPE, retinal pigment epithelium.

ii. As you suggested, we have shown the relative mRNA levels in Figure 7F. The mRNA levels of proangiogenic growth markers (*Vegfa*, *Pdgfb*, and *Tek*), an endothelial cell marker (*Tek*), and profibrotic factors (*Tgfb1* and *Pdgfb*) were higher in the CNV lesions of *Sts*^{-/-} mice than in those of WT mice (Figure 7F). However, the mRNA level of *Fgf2* was not changed (Figure R4).

Figure 7F

(F) Relative mRNA levels of proangiogenic growth markers (*Vegfa*, *Pdgfb*, and *Tek*), an endothelial cell marker (*Tek*), and profibrotic factors (*Tgfb1* and *Pdgfb*) in the RPE-choroid flat mounts of WT and *Sts*^{-/-} mice (n=4 per group).

Figure R4

Figure R4 There was no difference in the mRNA level of *Fgf2* between WT and *Sts*^{-/-} mouse (n=4 per group).

h. Figure S5

i. Retinal layers need to be labeled.

Response:

i. Naming of layers (INL, ONL, RPE, and Choroid) in the 3rd column within panel C&E were added. Thanks.

Figure S5

(C) Representative images of F4/80 (green) and YM1 (red) staining in retinal sections of WT and *Sts*^{-/-} mice.

(E) Representative images of F4/80 (green) and ARG1 (red) staining in retinal sections of WT and *Sts*^{-/-} mice.

Blue indicates DAPI. INL, inner nuclear layer; ONL, outer nuclear layer; RPE, retinal pigment epithelium.

i. Table S1

i. Did the authors design a Forward and Reverse primer for genotyping the *Sult2b1*^{-/-} and *Sts*^{-/-} mice? If so, this needs to be specified in the table.

Response:

i. Yes. We designed Forward and Reverse primers for genotyping the *Sult2b1*^{-/-} and *Sts*^{-/-} mice. We have specified them in the table.

Table S1 Genotyping primers.

Primer	Sequence (5'-3')
Sult2b1 - Forward primer	CTTATTCAACCACCACACCCAT
Sult2b1 - Reverse primer 1	TCCATCCCTAGCTTCACATGG
Sult2b1 - Reverse primer 2	GACAGCGCAGGGCCACAC
Sts -Forward primer	AGGGGGAAGGCCATGAGTACA
Sts -Reverse primer	ACTTCCTGTCCGTCTGACCTCAGT

Reference

1. Tobe T, Ortega S, Luna JD, Ozaki H, Okamoto N, Derevjanik NL, et al. Targeted disruption of the FGF2 gene does not prevent choroidal neovascularization in a murine model. *Am J Pathol*. 1998;153(5):1641-6.
2. Lambert V, Lecomte J, Hansen S, Blacher S, Gonzalez ML, Struman I, et al. Laser-induced choroidal neovascularization model to study age-related macular degeneration in mice. *Nat Protoc*. 2013;8(11):2197-211.
3. Shah RS, Soetikno BT, Lajko M, Fawzi AA. A Mouse Model for Laser-induced Choroidal Neovascularization. *J Vis Exp*. 2015(106):e53502.
4. Sene A, Khan AA, Cox D, Nakamura RE, Santeford A, Kim BM, et al. Impaired cholesterol efflux in senescent macrophages promotes age-related macular degeneration. *Cell Metab*. 2013;17(4):549-61.
5. Lin JB, Moolani HV, Sene A, Sidhu R, Kell P, Lin JB, et al. Macrophage microRNA-150 promotes pathological angiogenesis as seen in age-related macular degeneration. *JCI Insight*. 2018;3(7).
6. Yanai R, Mulki L, Hasegawa E, Takeuchi K, Sweigard H, Suzuki J, et al. Cytochrome P450-generated metabolites derived from ω -3 fatty acids attenuate neovascularization. *Proc Natl Acad Sci U S A*. 2014;111(26):9603-8.
7. Yanai R, Chen S, Uchi SH, Nanri T, Connor KM, Kimura K. Attenuation of choroidal neovascularization by dietary intake of ω -3 long-chain polyunsaturated fatty acids and lutein in mice. *PLoS One*. 2018;13(4):e0196037.
8. Fu Z, Liegl R, Wang Z, Gong Y, Liu CH, Sun Y, et al. Adiponectin Mediates Dietary Omega-3 Long-Chain Polyunsaturated Fatty Acid Protection Against Choroidal Neovascularization in Mice. *Invest Ophthalmol Vis Sci*. 2017;58(10):3862-70.
9. Koto T, Nagai N, Mochimaru H, Kurihara T, Izumi-Nagai K, Satofuka S, et al. Eicosapentaenoic acid is anti-inflammatory in preventing choroidal neovascularization in mice. *Invest Ophthalmol Vis Sci*. 2007;48(9):4328-34.
10. Zhu L, Parker M, Enemchukwu N, Shen M, Zhang G, Yan Q, et al. Combination of apolipoprotein-A-I/apolipoprotein-A-I binding protein and anti-VEGF treatment overcomes anti-VEGF resistance in choroidal neovascularization in mice. *Commun Biol*. 2020;3(1):386.
11. Terao R, Honjo M, Aihara M. Apolipoprotein M Inhibits Angiogenic and Inflammatory Response by Sphingosine 1-Phosphate on Retinal Pigment Epithelium Cells. *Int J Mol Sci*. 2017;19(1).
12. Yang S, Li T, Jia H, Gao M, Li Y, Wan X, et al. Targeting C3b/C4b and VEGF with a bispecific fusion protein optimized for neovascular age-related macular degeneration therapy. *Sci Transl Med*. 2022;14(647):eabj2177.
13. Zhao X, Gao M, Liang J, Chen Y, Wang Y, Wang Y, et al. SLC7A11 Reduces Laser-Induced Choroidal Neovascularization by Inhibiting RPE Ferroptosis and VEGF Production. *Front Cell Dev Biol*. 2021;9:639851.
14. Wu J, Chen J, Hu J, Yao M, Zhang M, Wan X, et al. CircRNA Uxs1/miR-335-5p/PGF axis regulates choroidal neovascularization via the mTOR/p70 S6k pathway. *Transl Res*. 2023;256:41-55.
15. Chen Y, Zhu X, Ye F, Wang H, Wan X, Zhang T, et al. Malondialdehyde-Modified Photoreceptor Outer Segments Promote Choroidal Neovascularization in Mice. *Transl Vis Sci Technol*. 2022;11(1):12.
16. Zhang P, Wang H, Luo X, Liu H, Lu B, Li T, et al. MicroRNA-155 Inhibits Polarization of Macrophages to M2-Type and Suppresses Choroidal Neovascularization. *Inflammation*. 2018;41(1):143-53.
17. Savla SR, Prabhavalkar KS, Bhatt LK. Liver X receptor: a potential target in the treatment of

atherosclerosis. *Expert Opin Ther Targets*. 2022;26(7):645-58.

18. Choudhary M, Ismail EN, Yao PL, Tayyari F, Radu RA, Nusinowitz S, et al. LXRs regulate features of age-related macular degeneration and may be a potential therapeutic target. *JCI Insight*. 2020;5(1).

19. Gnanaguru G, Wagschal A, Oh J, Saez-Torres KL, Li T, Temel RE, et al. Targeting of miR-33 ameliorates phenotypes linked to age-related macular degeneration. *Mol Ther*. 2021;29(7):2281-93.

20. Murray PJ. Macrophage Polarization. *Annual Review of Physiology*. 2017;79(1):541-66.

July 24, 2023

RE: Life Science Alliance Manuscript #LSA-2023-02020-TR

Dr. Xiaoling Wan
Shanghai General Hospital
100 Haining Road
Shanghai 200080
China

Dear Dr. Wan,

Thank you for submitting your revised manuscript entitled "Macrophage Sult2b1 promotes pathological neovascularization in age-related macular degeneration". We would be happy to publish your paper in Life Science Alliance pending final revisions necessary to meet our formatting guidelines.

- please add the Twitter handle of your host institute/organization as well as your own or/and one of the authors in our system
- please add callouts for Figure S6A-D to your main manuscript text
- you may want to consider uploading Figure 8 as a Graphical Abstract rather than as a figure, but this is up to you

A. FINAL FILES:

B. MANUSCRIPT ORGANIZATION AND FORMATTING:

****It is Life Science Alliance policy that if requested, original data images must be made available to the editors. Failure to provide**

original images upon request will result in unavoidable delays in publication. Please ensure that you have access to all original data images prior to final submission.**

The license to publish form must be signed before your manuscript can be sent to production. A link to the electronic license to publish form will be sent to the corresponding author only. Please take a moment to check your funder requirements.

Sincerely,

Reviewer #1 (Comments to the Authors (Required)):

The authors have answered all my questions and this manuscript is ready for publication.

July 31, 2023

RE: Life Science Alliance Manuscript #LSA-2023-02020-TRR

Dr. Xiaoling Wan
Shanghai General Hospital
100 Haining Road, Shanghai 200080, China
Shanghai 200080
China

Dear Dr. Wan,

Thank you for submitting your Research Article entitled "Macrophage Sult2b1 promotes pathological neovascularization in age-related macular degeneration". It is a pleasure to let you know that your manuscript is now accepted for publication in Life Science Alliance. Congratulations on this interesting work.

DISTRIBUTION OF MATERIALS:

Again, congratulations on a very nice paper. I hope you found the review process to be constructive and are pleased with how the manuscript was handled editorially. We look forward to future exciting submissions from your lab.

Sincerely,
